# Total delay and associated factors among tuberculosis patients in Jimma Zone, Southwest Ethiopia

Berhane Megerssa Ereso[1,2]*, Mette Sagbakken[3], Christoph Gradmann[1], Solomon Abebe Yimer[4,5]

1 Department of Community Medicine and Global Health, Institute of Health and Society, University of Oslo, Oslo, Norway, 2 Department of Health Policy and Management, Faculty of Public Health, Institute of Health, Jimma University, Jimma, Ethiopia, 3 Department of Nursing and Health Promotion, Faculty of Health Sciences, Oslo Met- Oslo Metropolitan University, Oslo, Norway, 4 Department of Microbiology, Institute of Clinical Medicine, University of Oslo, Oslo, Norway, 5 Coalition for Epidemic Preparedness Innovations (CEPI), Oslo, Norway

* berhanemegerssa2004@gmail.com

**Data Availability Statement:** All data analyzed during the present study are within the paper and its Supporting Information files.

**Funding:** This study was not funded by a grant. It is part of a PhD project and was supported by the

## Abstract

### Background

Delays in diagnosis and treatment of tuberculosis (TB) increases severity of illness and continued transmission of TB in the community. Understanding the magnitude and factors associated with total delay is imperative to expedite case detection and treatment of TB. The aim of this study was to determine the length and analyze factors associated with total delay.

### Methods

Analytic cross-sectional study was conducted in Jimma Zone, Southwest Ethiopia. All newly diagnosed TB patients > 15 years of age were included from randomly selected eight districts and one town in the study area. A structured questionnaire was applied to collect socio-demographic and clinical data. The median total delay was used to dichotomize the sample into delayed and non-delayed patient categories. Logistic regression analysis was used to analyse the association between independent and outcome variables. A p-value < 0.05 were considered statistically significant.

### Results

A total of 1,161 patients were included in this study. The median total delay was 35 days. Patients who had swelling or wound in the neck region were more likely to be delayed than their counterpart [adjusted odds ratio (AOR) = 3.02, 95% confidence interval (CI): 1.62, 5.62]. Women were more likely to experience longer total delay (AOR = 1.46, 95% CI:1.00, 2.14) compared to men. Patients who had poor knowledge of TB were more likely to be delayed compared to those who had good knowledge (AOR = 3.92, 95% CI: 2.65, 5.80).

Strategic and Collaborative Capacity Development in Ethiopia and Africa (SACCADE) Project, Norwegian Program for Capacity Development in Higher Education and Research for Development (NORHAD), University of Oslo. There was no additional external funding received for the current study

**Competing interests:** The authors have declared that no competing interests exist.

## Conclusion

The present study showed long total delay in diagnosis and treatment of TB. Targeted interventions that enhance TB knowledge and practice, expedite early suspect identification, referral and management of all forms of TB is imperative to reduce total delay in diagnosis and treatment of TB.

## 1. Introduction

Tuberculosis (TB) is a major cause of illness in low-resource countries. It is among the top 10 causes of death, and has been the leading cause of death from a single infectious agent in recent years [1, 2]. There were an estimated 10.0 million TB cases, and 1.5 million deaths due to TB in 2020 [3]. The 30 high TB burden countries are responsible for the majority (86%) of the estimated incident TB cases [2].

Ethiopia is among the high TB burden countries [3]. TB is a major public health problem and one of the leading infectious diseases in Ethiopia. There were an estimated 140 new incident TB cases per 100,000 population in 2020 and TB mortality rate of 19 per 100,000 populations in 2019 in Ethiopia [4]. Ethiopia accounts for 90,000 (3%) of annually missed TB cases worldwide [2, 5]. A recent national case detection rate of all forms of TB was 76 percent, which is below the target of World Health Organization [3, 6]. One of the contributing factors for low case detection is the delay in diagnosis and treatment of TB. Delay in diagnosis and treatment is commonly divided into three components (patients' delay, health system's delay and total delay). While patient delay refers to the delay period from onset of the major TB symptoms to first visit to a medical provider, health system delay encompasses the delay period from first visit to a medical provider to first start of anti-TB treatment. Total delay is defined as the delay period from start of major TB symptoms to first start of anti-TB treatment. Delay in TB diagnosis and treatment remains a major problem of TB control program generally in low and middle income countries [7] particularly in Ethiopia [8, 9]. Patients' health care seeking delay differs among different regions of Ethiopia [10, 11].

Prompt diagnosis and treatment is crucial for efficient TB control program performance and achieving the End TB targets. The target set for the End TB Strategy for the year 2035 include: 1) reduction of TB mortality by 95%, 2) decreasing TB incidence by 90% (i.e compared to the baseline of 2015), and 3) to certify that no family is suffered with TB related catastrophic costs [12]. The End TB Strategy targets can only be realized if diagnosis, treatment, and preventive services for TB are delivered based on the context of universal health coverage, which implies that all people with TB should be early detected and properly treated [2, 5]. Delayed diagnosis and treatment of TB cases has major role in the transmission of the disease in the community in most high TB burden countries. Early diagnosis and proper treatment of TB will reduce severity of illness, prevent transmission, increase cure rate, and prevent the development of drug-resistant TB [2, 13].

Various studies have been conducted on delays in diagnosis and treatment of TB in different parts of the world [14–21]. A systematic review and meta-analysis conducted in low- and middle-income countries showed that the median total delay ranged from 30 in Zimbabwe and Vietnam to 366.5 days in Afghanistan [7]. Another systematic review and meta-analysis conducted in Ethiopia revealed that the median diagnostic of 45days. The prevalence of diagnostic delay in Ethiopia ranged from 9.57% in Addis Ababa city and 68.84% in Somali region [22]. Various factors that are associated with diagnostic and treatment delay were identified in

earlier studies in Ethiopia and elsewhere. Some of these factors included poor knowledge about TB [23], patient first visit to lower level facilities [21], long distance to health facility [19], low level of income [23], being female [19, 24], being illiterate [15, 22], perceived TB stigma [25], rural residence [19, 22], having extra-pulmonary TB [14, 22, 26], having smear negative pulmonary TB [26], self-treatment [21, 23, 26], being HIV negative [22, 23], absence of chest pain and presence of haemoptysis [17] etc.

Although various studies were conducted in Ethiopia and elsewhere to assess TB diagnostic and treatment delay, there is limited study to date that has been conducted to assess the length and associated factors of total delay in diagnosis and treatment of TB in Jimma Zone, Ethiopia. The length of total delay and associated factors may vary according to the local setting, including the socio-demographic and economic condition of the population in the study area [10, 11]. Understanding the contribution of these factors is important to propose targeted interventions to address diagnostic and treatment delay at the local setting. Therefore, the aim of the present study was to determine the length and analyze factors associated with total delay in Jimma Zone, Ethiopia.

## 2. Materials and methods

### 2.1. Study setting and design

Analytic cross-sectional study was conducted among all forms of TB patients who started treatment from September 2016 to October 2017. The study was conducted in Jimma Zone, Oromia Region, Southwest Ethiopia. Jimma Zone is located 354 kilometres from Addis Ababa, the capital city of Ethiopia, with a total area of 199,316.18 square kilometres [Jimma Zone health office, 2016]. According to 2017 projected population census, Jimma Zone had an estimated population of 3,261,371, of which 49.9% were women [27]. In 2016, the Zone had 17 districts and two town administrations. A total of seven public hospitals (five were primary, one general and one specialized); 120 health centres, and 494 health posts were registered in the study area during the study period. The hospitals and health centres have been providing TB diagnostic and treatment services. Health extension workers at health posts have been rendering TB treatment services, screening and referring of TB suspects to the nearest health facilities for confirmatory testing. Non-governmental health facilities such as the Catholic mission and several private clinics were also providing TB diagnostic and treatment services. In 2016, a total of 3,008 all forms of TB patients were identified in Jimma Zone. Among these, 1,468 patients were bacteriologically confirmed pulmonary TB cases [Jimma Zone health office, 2016; Jimma town health office, 2016].

### 2.2. Study population and sampling

A total of 1,161 newly diagnosed TB patients were included from sampled districts' public health facilities and health posts. Patients whose age was less than 15 years, who could not respond to the interview questions and critically ill patients were excluded from the study. Eight districts and one town administration were selected from the 17 districts and two town administrations by using a simple random sampling method. Subsequently, all TB DOTS sites in the sampled districts and a town administration were covered by the study.

This study is part of a PhD project. It follows a recently published sub-study that compared community- versus facility-based DOTS in a cohort study that enrolled a total of 1,161 study participants [28]. We included all of the 1,161 study participants in the current study. However, in order to address the research objective in the current study, a total sample of 422 calculated using formula for estimating single proportion could have been enough (i.e. considering 50% proportion of delay of more than one month at 95% confidence interval and a margin

error of 5%). The inclusion of the 1,161 study participants in the current study is advantageous as it provides more than adequate representative population to address the research questions. The sample size was proportionally allocated to the selected public health facilities based on the previous one year's (before the study's start) patient flow. The study participants were consecutively included until the required sample size was achieved [Fig 1].

## 2.3. Data collection and analysis

A structured questionnaire was developed based on the national and WHO's guidelines; and previous studies [29–34]. The questionnaire was translated to the local language (Afan Oromo) by a professional whose mother tongue is the local language. It was checked and peer reviewed for any inconsistencies between the translated version and the original English version of the questions. The translated version of the questionnaire was re-translated to English by another professional who fluently spoke and wrote the local language. Then, it was pretested to check for the clarity and applicability of the questionnaire for the context of the study area. Based on the findings of the pre-test, modifications such as clarifying statements were

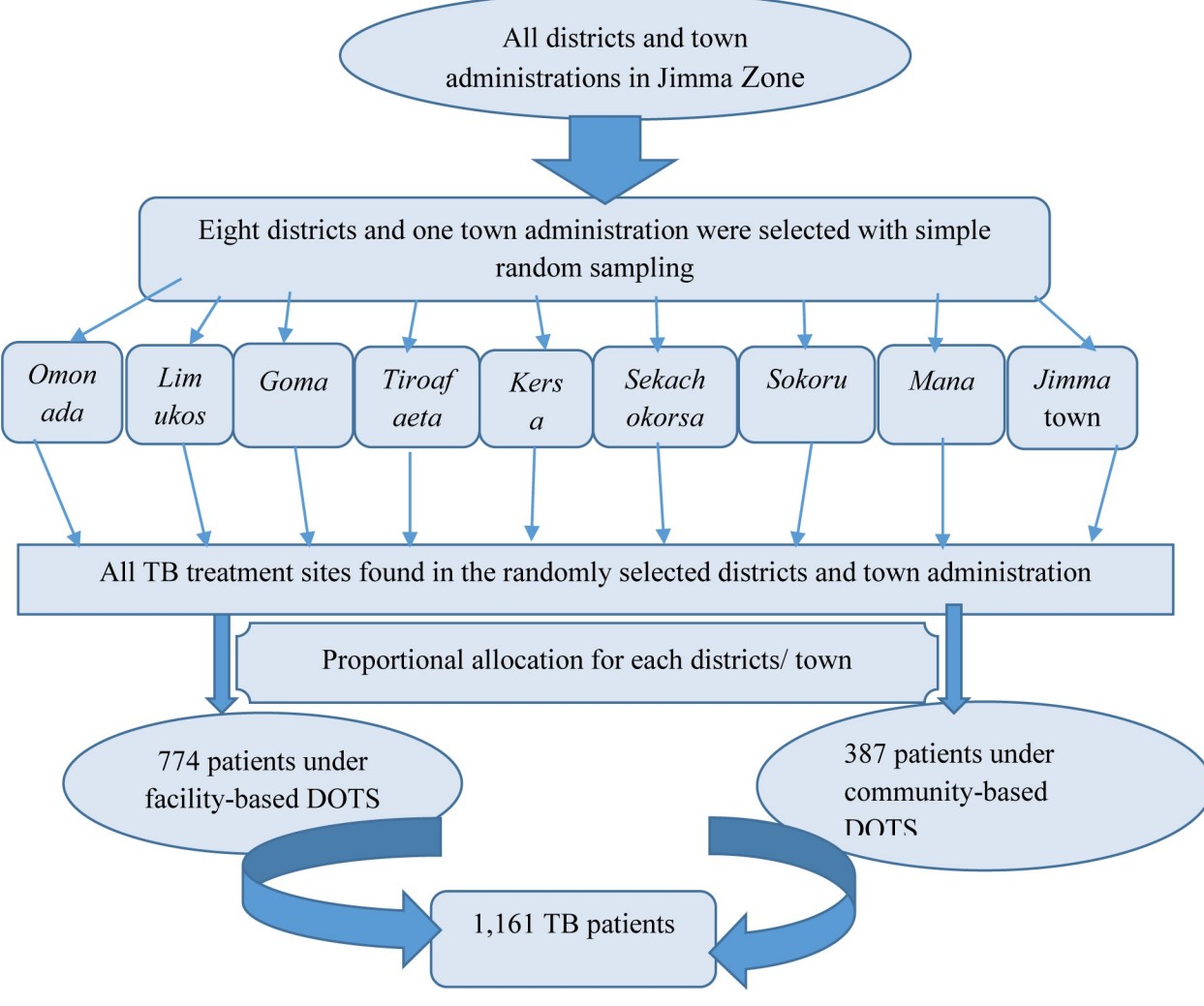

**Fig 1. Schematic presentation of sampling procedure for TB patients, Jimma Zone, 2017.**

made. We recruited experienced data collectors and supervisors and they were provided with the required training on the data collection and supervision techniques. The whole process of data collection was supervised by the principal investigator. The study participants were consecutively enrolled starting from September 2016 to October 2017. They were all interviewed during the enrolment time. Clinical data such as HIV status, main mode of TB diagnosis, TB classification were collected by a checklist attached to the questionnaire from TB register.

The collected data were checked for completeness and consistency, coded and entered into the EpiData entry client software, version 4.4.3.1. Then, the data were exported to the statistical package for social sciences software (SPSS) version 21 for analysis. Descriptive statistics were computed for the variables. To assess TB knowledge, a score of one was given for the correct responses and a score of zero was given for wrong responses. Then the total knowledge score and median score were computed. Those with a total score of less than the median value were categorized as having poor knowledge, while those who scored greater than or equal to the median value were classified as having good TB knowledge. Likewise, the total perceived stigma and median scores were determined. Those with a total score of less than the median value were classified as not having perceived stigma, whereas those who scored greater than or equal to the median value were categorized as having perceived stigma.

The median total delay was used to categorize the sample into delayed and non-delayed patient groups. To analyse the association of independent variables with the dependent variable, we used binary logistic regression analysis. First, bivariable analysis was performed for each independent variable against the respective outcome variable and crude odds ratio (COR) was calculated. Multivariable analysis was conducted comprising candidate variables in the bivariable analysis that scored a p-value of $< 0.25$ with a backward stepwise method. Besides, the respective adjusted odds ratios (AOR) and 95% confidence intervals (CI) were computed. A p-value $< 0.05$ was considered statistically significant.

## 2.4. Operational definitions

Patient delay: the time from onset of symptoms (cough) until first visit to a medical provider.

Health system delay: the time between the first visit to a medical provider and the first start of anti-TB treatment.

Total delay: the period between onset of TB symptoms and the first start of anti-TB treatment (the sum of patient delay and health system delay).

Delayed: operationally defined as delayed if the period between the onset of TB symptoms and first start of anti-TB treatment is more than the calculated median total delay ($> 35$ days).

Community based DOTS: TB treatment offered at a health post or patient's residence by a HEW or a trained TB treatment supporter.

Facility based DOTS: TB treatment offered at public health facility (health center or hospital) by a trained health care provider.

Medical provider/ formal health care provider: include hospitals, health centres, health posts, private clinics, and drug retail outlets.

Non formal-health care providers: traditional healers or herbalists, religious healers.

**2.4.1. TB diagnosis.** The national guideline for TB, and DR-TB was followed to diagnosis and treat TB [5]. Health care providers identify, triage and examine individuals who present with persistent cough of two or more weeks (any duration for HIV positive), fever for more than two weeks, night sweats, and weight loss of more than 1.5 kg per month. TB diagnosis is made with proper investigations using one or more of the following methods: mycobacteriological examination, chest x-ray, and cytological/histopathology (analysis of body parts/fluids) examinations. Acid-fast-bacilli (AFB) smear microscopy is the most common method used for

TB diagnosis and follow up of treatment response. A patient is diagnosed as a pulmonary positive TB case, when he/she has at least one positive result on AFB microscopy, or when his/ her Xpert MTB/RIF test result is positive for mycobacteria. A patient is diagnosed as a pulmonary negative TB case when he/she has signs and symptoms suggestive of TB with at least two negative result on AFB smear microscopy, and secondly, when his/her Xpert MTB/RIF test results detects mycobacterium and a decision to treat with a full course of anti-TB drugs is made. The decision is made based on suggestive findings from supporting laboratory tests and with the aid of proper clinical examination [5].

## 2.5. Ethical considerations

Ethical clearance was sought from the Regional Committee for Medical Research Ethics (REK Øst), Norway with reference number of 2015/2124 REK sør-øst B. Ethical approval was also obtained from the Institutional Review Board of Jimma University, Ethiopia with reference number of RPGC/389/2016. Permission was granted from Oromia regional health bureau, Ethiopia with reference number of BEFO/ABTF/1-8/2026 and Jimma zone health office, Ethiopia with reference number of WEFBJ/ 0-11/8060/08. Written (for literate) and oral (for illiterate) informed consent was secured from the study participants before starting the data collection. For minors (15 – 17years), assent was obtained from the study participants and consent was secured from their parents or guardians.

## 2.6. Inclusivity in global research

Additional information regarding the ethical, cultural and scientific considerations specific to inclusivity in global research is included in the S3 Text.

## 3. Results

### 3.1. Characteristics of the study participants

In this study, we assessed the length of total delay and associated factors among TB patients in Jimma Zone, Ethiopia. Accordingly, a total of 1,161 patients participated in this study: 774 (66.7%) under FB-DOTS, 387(33.3%) under CB-DOTS) were included. Of these, 51.2% were male, 65.9% were married, and 39.5% were illiterate. The mean (± SD) age for the study participants was 32.2 (±14.41) years, with a range of 15 to 90 years. Nearly half (46.9%) of the patients were 25–44 years of age. Of the total study participants, 63.7% were farmers by occupation, 51.7% were rural residents, and 48.8% were smear-positive pulmonary TB patients (Table 1).

### 3.2. Study districts and time to reach the nearest medical provider

A majority (18.9%) of the study participants were from *goma* district. The mean (± SD) time taken to reach the nearest medical provider was 32.86 (±28.52) minutes, ranging from two to 180 minutes (Table 2).

### 3.3. Patients' first visit to medical providers

A total of 708 (60.98%) patients visited other medical providers (health posts and private clinics) before visiting the current health facility where they got their TB diagnosis and initiated anti-TB treatment. Of these, a majority (169) of the patients who first visited private clinic were not delayed. On the contrary, most (45) of the patients who visited health posts were delayed (Fig 2).

**Table 1. Socio-demographic and clinical characteristics of the study participants in Jimma Zone, 2017 (N = 1,161).**

| Variables | | Frequency | Percent (%) |
|---|---|---|---|
| Sex | Male | 594 | 51.2 |
| | Female | 567 | 48.8 |
| Age in year | 15–24 | 365 | 31.4 |
| | 25–44 | 544 | 46.9 |
| | 45–64 | 202 | 17.4 |
| | ≥65 | 50 | 4.3 |
| | Mean ± SD (32.22±14.41); Minimum 15; Maximum 90; Median 30 (IQR 22, 40) | | |
| Marital status | Single | 330 | 28.4 |
| | Married | 765 | 66.1 |
| | Divorced | 30 | 2.6 |
| | Widowed | 34 | 2.9 |
| Educational level | Illiterate | 459 | 39.5 |
| | Read and write only | 98 | 8.4 |
| | Primary school | 396 | 34.1 |
| | Secondary school | 132 | 11.4 |
| | College/University | 76 | 6.5 |
| Occupation | Farmer | 739 | 63.7 |
| | Merchant | 74 | 6.4 |
| | Government or NGO employee | 58 | 5.0 |
| | Daily laborer | 88 | 7.6 |
| | Housewife | 19 | 1.6 |
| | Student | 142 | 12.2 |
| | Unemployed | 41 | 3.5 |
| Religion | Orthodox Christian | 160 | 13.8 |
| | Muslim | 939 | 80.8 |
| | Protestant | 61 | 5.3 |
| | Catholic | 1 | 0.1 |
| Residence | Urban | 561 | 48.3 |
| | Rural | 600 | 51.7 |
| Source of household income | Farming | 769 | 66.2 |
| | Monthly salary | 59 | 5.1 |
| | Private/ trading | 78 | 6.7 |
| | Daily payment | 82 | 7.1 |
| | Family/relative | 152 | 13.1 |
| | Prefer not to answer | 21 | 1.8 |
| Monthly household income in ETB | ≤ 1000 | 584 | 50.3 |
| | 1001–2500 | 388 | 33.4 |
| | 2501–3500 | 41 | 3.5 |
| | > 3500 | 90 | 7.8 |
| | Do not have regular income | 58 | 5.0 |
| Previous contact with TB patient | Yes | 118 | 10.2 |
| | No | 807 | 69.5 |
| | Don't know /not sure | 236 | 20.3 |
| Cigarette smoking | Yes | 71 | 6.1 |
| | No | 1090 | 93.9 |
| **Variables** | | Frequency | Percent (%) |

*(Continued)*

**Table 1.** (Continued)

| Variables | | Frequency | Percent (%) |
|---|---|---|---|
| Drink alcohol | Yes | 93 | 8.0 |
| | No | 1068 | 92.0 |
| Khat chewing | Yes | 599 | 51.6 |
| | No | 562 | 48.4 |
| Previous Rx for TB | Yes | 33 | 2.8 |
| | No | 1128 | 97.2 |
| History of diabetes mellitus | Yes | 12 | 1.0 |
| | No | 1149 | 99.0 |
| Main source of information about TB | Health care providers | 914 | 78.7 |
| | TV/Radio | 172 | 14.8 |
| | Family/relative | 53 | 4.6 |
| | Others | 22 | 1.9 |
| Type of DOTS | Facility-based | 774 | 66.7 |
| | Community-based | 387 | 33.3 |
| TB classification | Smear-positive pulmonary TB | 567 | 48.8 |
| | Smear-negative pulmonary TB | 251 | 21.6 |
| | Extra pulmonary TB | 343 | 29.5 |
| Main mode of TB diagnosis | Bacteriological | 592 | 51.0 |
| | Histopathology/ Biopsy | 226 | 19.5 |
| | Radiological | 301 | 25.9 |
| | Clinical clues | 42 | 3.6 |
| HIV status | Reactive | 38 | 3.3 |
| | Non-reactive | 1040 | 89.6 |
| | Unknown | 83 | 7.1 |

## 3.4. Patient delay and total delay

The median patient delay (time from onset of symptoms until first visit to a medical provider) was 30 days [interquartile rang (IQR): 17.5, 60 days]. Whereas, the median total delay was 35 days (IQR: 25, 67 days). The total delay ranged from 4 to 732 days (Table 3).

**Table 2. Districts and time to reach the nearest medical provider of the study participants at Jimma Zone, 2017 (N = 1,161).**

| Variables | | Frequency | Percent (%) |
|---|---|---|---|
| District / town administration | *Goma* | 220 | 18.9 |
| | *Jimma* | 157 | 13.5 |
| | *Kersa* | 122 | 10.5 |
| | *Limu kosa* | 144 | 12.4 |
| | *Mana* | 97 | 8.4 |
| | *Omo nada* | 102 | 8.8 |
| | *Seka chokorsa* | 120 | 10.3 |
| | *Sokoru* | 108 | 9.3 |
| | *Tiro afeta* | 91 | 7.8 |
| Time to reach nearest medical provider in minutes | ≤ 30 | 792 | 68.2 |
| | 31–60 | 278 | 23.9 |
| | > 60 | 91 | 7.9 |
| | Mean ± SD (32.86±28.52); Minimum 2; Maximum 180; Median 25 (IQR 15, 45) | | |

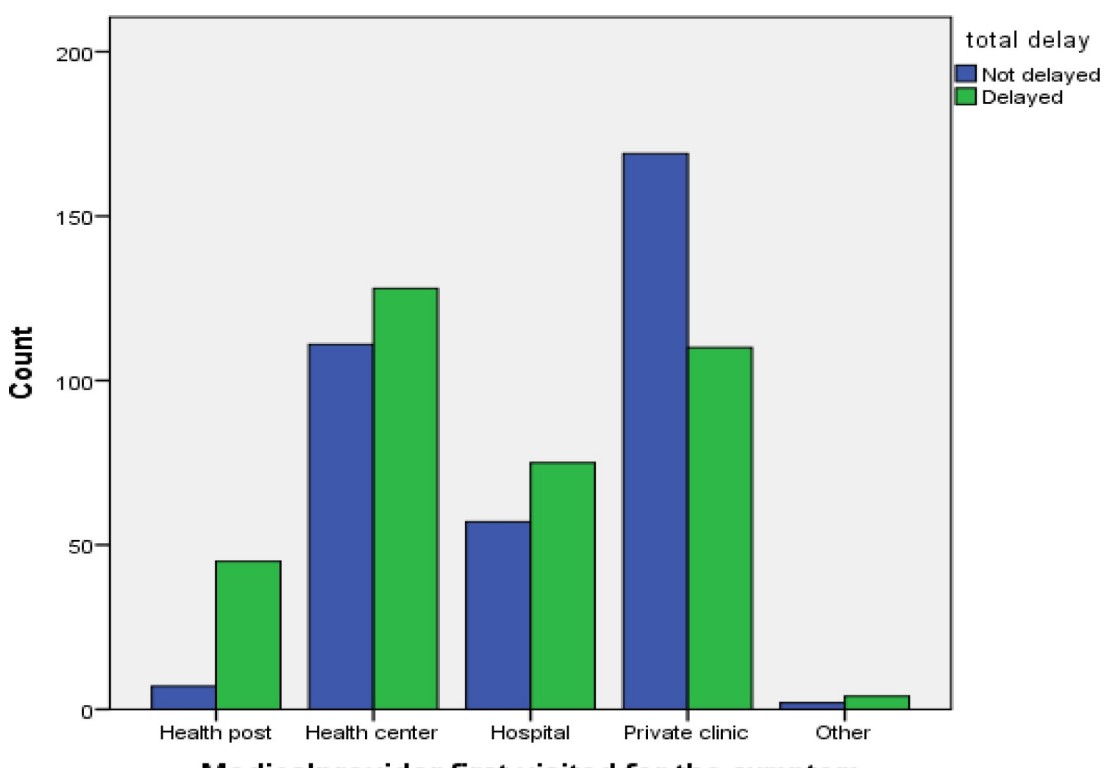

**Fig 2. Medical provider first visited for the symptoms with patient delay status at Jimma Zone, 2017.**

### 3.5. Determinants of total delay

The median total delay was 35(IQR 25, 67 days) days, and for more than half (586) of the patients, the total delay was > 35 days. Variables such as occupation, source of household income, being a smoker, alcohol use, khat chewing, HIV status, TB classification, chest pain, and weight loss were removed from the final model even if these were statistically significant during the bivariate analysis. In the multivariate analysis, patients who had poor knowledge about TB were about four times more likely to be delayed than patients who had good knowledge about TB (AOR = 3.92, 95% CI: 2.65, 5.80). Those patients who had swelling or wound in the neck region were three times more likely to be delayed than their counterpart (AOR = 3.02, 95% CI: 1.62, 5.62). Patients who attended college or university were 72% less likely to be delayed than those patients who did not read and write (AOR = 0.28, 95% CI: 0.10, 0.81). Female patients were about 1.5 times more likely to be delayed than their counterparts (AOR = 1.46, 95% CI: 1.00, 2.14). Patients who had a monthly household income of 1001–2500 ETB had increased total delay compared to patients who earned > 3500 ETB

**Table 3. Patient, health system and total delays for the study participants in Jimma Zone, 2017.**

| Variable | Median | IQR | Minimum | Maximum |
|---|---|---|---|---|
| Duration from onset of symptoms until first visit to a medical provider in days | 30.00 | 17.5, 60 | 2 | 730 |
| Duration from first visit to a medical provider until diagnosis of TB in days | 3.00 | 2, 9 | 1 | 365 |
| Duration from diagnoses of TB until treatment started in days | 1.00 | 1, 2 | 1 | 90 |
| Total time taken from start of TB symptoms until start of treatment | 35.00 | 25, 67 | 4 | 732 |

(AOR = 15.75, 95% CI: 2.92, 84.91). Those patients who traveled 30 or less minutes to reach the nearest medical provider were 53% less likely to be delayed than those patients who traveled more than 60 minutes (AOR = 0.47, 95% CI: 0.23, 0.94). Patients under CB-DOTS were about 1.5 times more likely to be delayed than their counterparts (AOR = 1.53, 95% CI: 1.02, 2.29). Patients who had no limitation of their day-to-day activity were less likely to be delayed than those patients who had high limitation of their day-to-day activity (AOR = 0.09, 95% CI: 0.01, 0.89) (Table 4).

## 4. Discussion

In this study, we assessed the length and associated factors of total delay among TB patients. A systematic review and meta-analysis conducted in low- and middle-income countries reported that the median total delay ranged from 30 to 366.5 days [7]. We observed a median total delay of 35 days which is similar to the study done in Addis Ababa, Ethiopia [35], but lower than the findings reported from earlier studies conducted in different Regions of Ethiopia [23, 25, 36–38]. The shorter total delay observed in the present study may be related to several reasons: these may include improved access to TB diagnostic and treatment facilities through decentralization of health services to the community, and health extension workers' contribution in identifying suspected TB cases and referring them to health centers for smear microscopy test [39].

Even though the observed total delay in our study was relatively lower compared to earlier study findings in Ethiopia, it is still long delay given the need for immediate diagnosis and treatment initiation for TB patients. In a recent qualitative study conducted in the study area, different barriers were identified that may have implications for increased total delay. Among these were shortage of resources including basic infrastructure and limited TB diagnostic services; and delays in the diagnostic process at the health facilities [40]. Solving these barriers through targeted interventions is crucial to reduce the observed relatively long total delay. Long delays in diagnosis and treatment initiation increase the severity and complications of disease that may result in unfavorable treatment outcome [37, 41]. It also increases the risk of developing drug resistant TB which leads to treatment failure, high mortality rates, and high transmission rate of drug resistant TB [42, 43]. These findings emphasize the importance of early diagnosis and prompt treatment of TB. Conversely, the observed median total delay in the current study is higher than the study findings in other parts of Ethiopia that reported 23 days and 33 days [8, 9]. This discrepancy might be because of differences in the study settings i.e. socio-economic status, study population, access to TB diagnostic facilities, and utilization of health care services among the population.

The analysis of socio-demographic, economic, and clinical factors revealed significant associations with increased total delay. In line with our study, low level of education [7, 25, 36, 38], long distance [19, 36], and poor knowledge about TB [23, 36, 37] were identified as predictors of total delay in former studies in Ethiopia and elsewhere. Our findings underscored that educational level is an important determinant of total delay. A former study in Ethiopia revealed that individuals with a higher level of education had better knowledge about TB compared to those who did not read and write [44]. Our study showed that the educational level of the patients may not be directly associated with the delay but do significantly affect the knowledge level of the study participants. A study from the Gambia showed an association of educational level with knowledge, attitude and health-seeking behavior regarding TB [45]. Another study from Malaysia reported that TB education program at school was an effective intervention for improvement in the mean score of knowledge, preventive practice, and perceived stigma about TB [46]. In addition, a study conducted in Ethiopia indicated that there was association

**Table 4. Determinents of total delay among the study partcipants at Jimma Zone, 2017.**

| Variables | | Total delay, No (%) | | COR (95% CI) | AOR (95% CI) | P-values |
|---|---|---|---|---|---|---|
| | | Not delayed ≤ 35 days (median) | Delayed > 35 days (median) | | | |
| Sex | Male | 297 (51.7) | 297 (50.7) | 1 | 1 | |
| | Female | 278 (48.3) | 289 (49.3) | 0.96 [0.76,1.21] | 1.46 [1.00,2.14] | 0.050 |
| Age groups in years | 15–24 | 185 (32.2) | 180 (30.7) | 1 | 1 | |
| | 25–44 | 269 (46.8) | 275 (46.9) | 0.60 [0.33,1.09] | 0.59 [0.20,1.72] | 0.331 |
| | 45–64 | 102 (17.7) | 100 (17.1) | 0.63 [0.35,1.14] | 0.53 [0.19,1.47] | 0.220 |
| | ≥ 65 | 19 (3.3) | 31 (5.3) | 0.60 [0.32,1.13] | 0.49 [0.17,1.39] | 0.181 |
| Educational level | Illiterate | 177 (30.8) | 282 (48.1) | 1 | 1 | |
| | Read and write only | 63 (11.0) | 35 (6.1) | 1.97 [1.21,3.21] | 0.74 [0.25,2.19] | 0.590 |
| | Primary school | 212 (36.9) | 184 (31.4) | 0.69 [0.37,1.27] | 0.41 [0.13,1.27] | 0.122 |
| | Secondary school | 81 (14.1) | 51 (8.7) | 1.07 [0.66,1.76] | 0.36 [0.13,1.02] | 0.053 |
| | College/University | 42 (7.3) | 34 (5.8) | 0.78 [0.44,1.38] | 0.28 [0.10,0.81] | 0.019 |
| Occupation | Farmer | 365 (63.5) | 374 (63.8) | 1 | 1 | |
| | Merchant | 37 (6.4) | 37 (6.3) | 0.48 [0.24,0.93] | 0.84 [0.18,3.97] | 0.820 |
| | Go/NGO employee | 33 (5.7) | 25 (4.3) | 0.46 [0.21,1.03] | 0.38 [0.02,7.29] | 0.523 |
| | Daily labourer | 33 (5.7) | 55 (9.4) | 0.35 [0.15,0.81] | 1.351 [0.07,27.31] | 0.845 |
| | House wife | 7 (1.2) | 12 (2.0) | 0.77 [0.35,1.69] | 0.56 [0.05,6.78] | 0.652 |
| | Student | 87 (15.1) | 55 (9.4) | 0.80 [0.25,2.49] | 1.44 [0.19,10.69] | 0.722 |
| | Unemployed | 13 (2.3) | 28 (4.8) | 0.29 [0.14,0.62] | 0.61 [0.16, 2.27] | 0.457 |
| Source of Household income | Farming | 380 (66.1) | 389 (66.4) | 1 | 1 | |
| | Monthly salary | 36 (6.3) | 23 (3.9) | 6.14 [1.80,21.02] | 0.17 [0.02,1.54] | 0.114 |
| | Private/trading | 40 (7.1) | 38 (6.5) | 3.83 [1.01,14.49] | 0.14 [0.01,1.54] | 0.107 |
| | Daily payment | 28 (4.9) | 54 (9.2) | 5.70 [1.55,20.92] | 0.37 [0.04,3.58] | 0.387 |
| | Family/relative | 73 (12.7) | 79 (13.5) | 11.57 [3.14,42.66] | 0.50 [0.05,4.95] | 0.552 |
| | Prefer not to answer | 18 (3.1) | 3 (0.5) | 6.49 [1.84,22.96] | 0.39 [0.05,3.31] | 0.389 |
| Monthly Household income in ETB | ≤ 1000 | 301 (52.3) | 283 (48.3) | 5.88 [2.74,12.61] | 9.51 [1.82,49.72] | 0.008 |
| | 1001–2500 | 162 (28.2) | 226 (38.6) | 8.72 [4.02,18.89] | 15.75 [2.92,84.91] | 0.001 |
| | 2501–3500 | 23 (4.0) | 18 (3.1) | 4.89 [1.86,12.88] | 6.92 [1.03,46.58] | 0.047 |
| | > 3500 | 39 (6.8) | 51 (8.7) | 1 | 1 | |
| | No regular income | 50 (8.7) | 8 (1.4) | 8.17 [3.48,19.22] | 13.29 [2.27,77.65] | 0.004 |
| Time to reach nearest medical provider in minute | ≤ 30 | 391 (68.0) | 401 (68.4) | 0.77 [0.49,1.19] | 0.47 [0.23,0.94] | 0.034 |
| | 31–60 | 145 (25.2) | 133 (22.7) | 0.69 [0.43,1.12] | 0.68 [0.31,1.46] | 0.317 |
| | > 60 | 39 (6.8) | 52 (8.9) | 1 | 1 | |
| Cigarette smoking | Yes | 25 (4.3) | 46 (7.8) | 1.87 [1.14,3.09] | 1.22 [0.54,2.75] | 0.627 |
| | No | 550 (95.7) | 540 (92.2) | 1 | 1 | |
| Alcohol use | Yes | 36 (6.3) | 57 (9.7) | 1.61 [1.05,2.49] | 1.01 [0.49,2.08] | 0.985 |
| | No | 539 (93.7) | 529 (90.3) | 1 | 1 | |

*(Continued)*

**Table 4.** (Continued)

| | | | | COR (95% CI) | AOR (95% CI) | P-values |
|---|---|---|---|---|---|---|
| Khat chewing | Yes | 320 (55.7) | 279 (47.6) | 0.72 [0.58,0.91] | 0.83 [0.56,1.24] | 0.356 |
| | No | 255 (44.3) | 307(52.4) | 1 | 1 | |
| HIV status | Reactive | 20 (3.5) | 18 (3.1) | 1 | 1 | |
| | Non-reactive | 525 (91.3) | 515 (87.9) | 0.51[0.23,1.11] | 0.98 [0.28,3.45] | 0.970 |
| | Unknown | 30 (5.2) | 53 (9.0) | 0.56 [0.35,0.88] | 0.96 [0.47,1.98] | 0.917 |
| Current illness status | No limitation of day-to-day activity | 362 (63.0) | 229 (39.1) | 0.25 [0.05,1.31] | 0.09 [0.01,0.89] | 0.040 |
| | Slight limitation | 211(36.7) | 352(60.0) | 0.67 [0.13,3.47] | 0.22 [0.02,2.12] | 0.189 |
| | High limitation | 2(0.3) | 5(0.9) | 1 | 1 | |

| Variables | | Total delay, No (%) | | COR (95% CI) | AOR (95% CI) | P-values |
|---|---|---|---|---|---|---|
| | | Not delayed ≤ 35 days (median) | Delayed > 35 days (median) | | | |
| Type of DOTS | Facility-based | 360 (62.6) | 414 (70.6) | 1 | 1 | |
| | Community-based | 215 (37.4) | 172 (29.4) | 1.44 [1.13,1.84] | 1.53 [1.02,2.29] | 0.041 |
| TB classification | Smear-positive PTB | 306 (53.2) | 221 (44.5) | 1 | 1 | |
| | Smear-negative PTB | 131(22.8) | 120 (20.5) | 0.57 [0.44,0.75] | 0.78 [0.48,1.26] | 0.309 |
| | Extra PTB | 138(24.0) | 205(35.0) | 0.62 [0.44,0.86] | 1.07 [0.64,1.79] | 0.798 |
| Medical provider first visited for symptoms | Health post | 7 (2.0) | 45 (12.4) | 1 | 1 | |
| | Health center | 111(32.1) | 128 (35.4) | 3.21 [0.49,20.96] | 2.23 [0.25,19.50] | 0.470 |
| | Hospital | 57 (16.5) | 75 (20.7) | 0.58 [0.10,3.21] | 0.79 [0.12,5.86] | 0.814 |
| | Private clinic | 169 (48.8) | 110(30.4) | 0.66 [0.12,3.72] | 0.49 [0.07,3.77] | 0.499 |
| | Other | 2 (0.6) | 4 (1.1) | 0.33 [0.06,1.81] | 0.39 [0.05,2.89] | 0.356 |
| Chest pain | Yes | 395 (68.7) | 328 (56.0) | 1 | 1 | |
| | No | 180 (31.3) | 258 (44.0) | 0.58 [0.46,0.74] | 0.75 [0.50,1.13] | 0.172 |
| Haemoptysis | Yes | 83 (14.4) | 75 (12.8) | 0.87 [0.62,1.22] | 1.24 [0.68,2.27] | 0.479 |
| | No | 492 (85.6) | 511 (87.2) | 1 | 1 | |
| Weight loss (10%) | Yes | 372 (64.7) | 345 (58.9) | 0.78 [0.62,0.99] | 1.28 [0.86,1.91] | 0.217 |
| | No | 203 (35.3) | 241(41.1) | 1 | 1 | |
| Swelling or wound around neck region | Yes | 42 (7.3) | 106(18.1) | 2.80[1.92,4.09] | 3.02 [1.62,5.62] | < 0.001 |
| | No | 533 (92.7) | 480(81.9) | 1 | 1 | |
| Knowledge about TB | Poor | 130 (22.6) | 333 (56.8) | 4.51[3.49,5.81] | 3.92 [2.65,5.80] | < 0.001 |
| | Good | 445 (77.4) | 253 (43.2) | 1 | 1 | |
| Perceived to be stigmatized | No | 183 (31.8) | 183 (31.2) | 0.97 [0.76,1.25] | 1.40 [0.89,2.20] | 0.143 |
| | Yes | 392 (68.2) | 403 (68.8) | 1 | 1 | |

AOR = adjusted odds ratio; COR = crude odds ratio; CI = confidence interval. 1 = Reference group

between educational level and knowledge about TB in which having good knowledge of TB led to a more positive attitude and better practice in relation to TB prevention and control [47]. This was supported by findings from a qualitative meta-synthesis conducted in Nigeria which reported that low education level and poor knowledge about TB were major barriers to TB diagnosis and treatment [20]. Compared to the uneducated persons, individuals with better educational level are more likely to have access to better economic situation which consecutively improves their access to health information and care [8, 44]. Moreover, the level of their knowledge about TB influence their perceptions about TB and change their health care-seeking behaviour [8, 44].

We observed a strong association between having swelling or wound in the neck region and long total delay. Cervical TB lymphadenitis is one of the most common manifestations of

extra-pulmonary TB which contributes to diagnostic delay. The common sign and symptom is a chronic, painless swelling in the neck region (no other remarkable symptom). Thus, the diagnosis of this type of TB requires a high index of suspicion and use of different diagnostic methods. Practically, it is not possible to apply all diagnostic procedures for all symptomatic patients. In this regard, ruling out for other causes of neck mass could contribute to diagnostic delay. Because of the absence of culture and pathology services in most health centers in Ethiopia including the study area, it is difficult to early diagnose such type of TB [48, 49]. A study from Zanzibar showed that from all patients with long total delay (> 6 months), a majority (90%) were patients with TB lymphadenitis [50]. Extra-pulmonary TB cases are seldom infectious as they are considered to have less contribution in TB transmission. However, delay in starting treatment for the patients may result in disseminated TB and increased mortality [49, 50]. As peripheral health care facilities in the study area do not have the capacity to diagnose extra-pulmonary TB cases, it is important to develop diagnostic algorithms for timely referral of presumptive extra-pulmonary TB patients. Therefore, training is essential for the health extension workers so that they can screen TB lymphadenitis suspects in the community for prompt referral to the next level of health care for better diagnosis and treatment [51].

Our study revealed that women were more likely to have long total delay compared to men. This finding is consistent with former studies conducted in Ethiopia and elsewhere [36, 52, 53]. Economic status, cultural beliefs, and perceived stigma are barriers of early care seeking for most female patients [54]. In terms of cultural beliefs, women in Ethiopia give priority for their family than themselves, have main responsibility for children care, have less decision making power, and worries about preserving modesty might affect their health care seeking behavior [55]. Previous study revealed that women were more likely to suffer from TB stigma compared to men [56]. Women in high burden countries like Ethiopia experienced long delays in TB diagnosis and treatment because of challenges related to TB services [57]. Most women faced problems in accessing TB service because of resource limitations, power imbalances, and poor knowledge about TB [57]. Women who have less access to quality healthcare, and delayed to seek formal health care were prone to long delays [54]. This may partly be related to the fact that women have the main responsibility for domestic tasks and care for their family members, especially children and elders. Women often lack economic independence, have less time and are less empowered compared to men in regards to care for themselves [36]. Therefore, it is important to recognize sex differences in individual's TB care-seeking. Targeted interventions that enhance women empowerment are required to improve poor health outcomes among women. Socio-economic support interventions could change their health seeking behaviour, improve adherence and treatment outcome for TB patients who are women [58, 59]. In addition, gender-based health education and behavioural change communication (BCC) interventions focusing on early health seeking counselling are required to reduce the burden of TB among women. BCC strategies are efficient, effective, sustainable and acceptable because messages can tailored to gender- or age-based target groups [60, 61].

Monthly household income was another determinant variable that was significantly associated with total delay. This finding is in line with previous study results from Ethiopia [8, 62, 63] and Pakistan [18]. This could be due to the fact that individuals who did not have adequate monthly income could not be able to seek health care early because of several reasons such as unaffordable medical costs and transport expenses [8, 63]. Therefore, they are likely to be prone to delayed diagnosis and treatment start. In resource limited settings like Ethiopia where there are several traditional practices and low access to quality health service, patients commonly seek health care from informal medical providers, as a result the patients might get traditional treatment that could delay them from timely seeking medical care from formal medical providers. TB diagnosis and treatment services in Ethiopia is given for free, money is

mostly needed for covering costs for transportation, accommodation and food for the patient. Individuals who have low level of income may need to work long hours, as a result they may not have time to early seek health care when they get ill. Patients with inadequate income are more likely to visit formal-health care provider only when they become critically ill [8, 63]. A systematic review conducted in high TB burden settings reported that livelihood, work, and family were given priority which led them to have a long delay in health care-seeking [57]. This finding shows the importance of improving the socio-economic condition of the population for effective TB control. As per the Sustainable Development Goals (SDG), the world is committed to end poverty by 2030 [64, 65]. According to a recent World Bank report, the poverty rate in Ethiopia fell from 44 percent in 2000 to 21 percent in 2018 [66, 67]. If this trend continues, the contribution of poverty to TB disease and transmission may be significantly reduced in Ethiopia including the study area.

Time taken to reach the nearest medical provider was associated with total delay. Long distance traveling to reach the nearest health facility was also reported in previous studies conducted in Ethiopia and other countries as barrier to individuals' health-seeking behavior which leads to lengthening of patient's delay [17, 68–72]. A recent qualitative study from the study area identified that long distance traveling to get TB diagnostic services was a barrier for most TB patients from the rural settings [40]. This could also be related with patients who were under FB-DOTS travel longer distance and more delayed than those patients under FB-DOTS.

Our study showed that patients under CB-DOTS were more likely to be delayed than those patients under FB-DOTS. This could be due to the fact that patients under CB-DOTS were rural residents and most of patients under FB-DOTS were urban residents. Rural residence was reported as independent predictor of delay in a systematic review conducted in Ethiopia [11]. One of the reasons might be that rural residents had poor access to health information and TB diagnostic and treatment facilities compared to urban residents. This might be due to the scarcity of formal health care providers in rural area of Ethiopia. Thus, patients from rural areas might take long chains of care-seeking through informal health care providers [57, 73]. A previous study from Southwest Ethiopia revealed that proportion of households sought care from formal health care providers was lower among rural compared to urban households [74]. Moreover, rural residents lack regular health information about the disease and do not seek health care early compared to the urban residents [10, 11]. In addition, rural residency makes it hard to travel to diagnostic and treatment facilities due to absence of road or distance (as long distance travelling discourages initiation of tuberculosis treatment) [10, 11].

This study has strengths and limitations. The strength of the study is that we included a relatively larger sample size compared to sample sizes used in other similar studies in Ethiopia. Thus, it provides more than adequate representative population to address the research questions. With regards to potential limitations, the study was only carried out in government health facilities; hence the findings cannot be generalized to all TB patients in the study area. There are also several private health care facilities and non-formal health care providers whereby TB patients may seek health care for their symptoms. Moreover, the present study focused on adult TB patients aged > 15 years who were treated at public health facilities; therefore, the results among other age groups and in other similar settings during the study period might be different. In addition, the reported duration of symptoms and first visit for treatment seeking is based on patients' ability to recall and interpretation of their symptoms. Since patients might not exactly remember the exact date of start of their symptoms and first visit to a medical provider, it is subject to a recall bias. However, we have tried to reduce the recall bias by using different techniques such as using local calendars linking to major religious and

national days to explain their perceived date of onset of TB symptoms and first visit to a medi-cal provider.

## 5. Conclusion

The present study showed long total delay (the median total delay of 35 days) in diagnosis and treatment start for TB patients in the study area. The study also revealed important factors associated with total delay. Poor knowledge of TB, swelling or wound around the neck region, being women and low level of education were identified as factors associated with total delay. Several interventions are required to reduce the observed length of total delay. Efficient imple-mentation strategies for early case detection and treatment initiation should be practiced to shorten total delay. Among others, well-crafted IEC and BCC strategies on TB need to be designed to increase awareness and health service utilization among the population in the study area. Swelling or wound around the neck region in a TB patient may be linked to TB lymphadenitis or disseminated TB. The health care facilities at peripheral level in the study area do not have the capacity to diagnose TB lymphadenitis cases. Thus, it is important to develop diagnostic algorithms for timely detection and referral of presumptive TB lymphade-nitis patients to the nearest medical providers with the capacity to diagnose TB lymphadenitis. Then, the delay period among this group of patients is shortened and prompt treatment is ini-tiated on time.

## Supporting information

**S1 Text. Questionnaire and information sheet with consent form English and local lan-guage versions.**
(PDF)

**S2 Text. STROB checklist.**
(PDF)

**S3 Text. PLOS' questionnaire on inclusivity in global research.**
(PDF)

**S1 Data. Data for current study with SPSS.**
(SAV)

## Acknowledgments

Our gratitude goes to the University of Oslo, Institute of Health and Society and Jimma Uni-versity for supporting us to undertake this study. We would like to thank the Oromia Health Bureau, the Jimma Zone and Jimma Town health offices and all the study participants for pro-viding us with the necessary information.

## Author Contributions

**Conceptualization:** Berhane Megerssa Ereso, Mette Sagbakken, Solomon Abebe Yimer.

**Data curation:** Berhane Megerssa Ereso.

**Formal analysis:** Berhane Megerssa Ereso, Solomon Abebe Yimer.

**Investigation:** Berhane Megerssa Ereso.

**Methodology:** Berhane Megerssa Ereso, Mette Sagbakken, Christoph Gradmann, Solomon Abebe Yimer.

**Resources:** Berhane Megerssa Ereso.

**Supervision:** Mette Sagbakken, Christoph Gradmann, Solomon Abebe Yimer.

**Visualization:** Berhane Megerssa Ereso, Mette Sagbakken, Solomon Abebe Yimer.

**Writing – original draft:** Berhane Megerssa Ereso.

**Writing – review & editing:** Berhane Megerssa Ereso, Mette Sagbakken,
Christoph Gradmann, Solomon Abebe Yimer.

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
