## [Decision Letter · Decision Letter 0]

16 Aug 2022

PONE-D-22-02817Total delay and associated factors among tuberculosis patients in Jimma Zone, Southwest EthiopiaPLOS ONE

Dear Dr. Ereso,

Thank you for submitting your manuscript to PLOS ONE. After careful consideration, we feel that it has merit but does not fully meet PLOS ONE’s publication criteria as it currently stands. Therefore, we invite you to submit a revised version of the manuscript that addresses the points raised during the review process.

We look forward to receiving your revised manuscript.

Kind regards,

Wenhui Mao, PhD

Academic Editor

PLOS ONE

Journal Requirements:

2. We note that this work is related to the following submission of yours currently under review: PONE-D-22-05873 "Determinants of unfavorable treatment outcome among tuberculosis patients in Jimma Zone, Southwest Ethiopia". Please ensure you have discussed the related submission in your cover letter. Please also revise your Baseline Demographics table to ensure that only clinical characteristics relevant to the current study are included, to avoid inappropriate overlap between the two works.

3. You indicated that you had ethical approval for your study. In your Methods section, please ensure you have also stated whether you obtained consent from parents or guardians of the minors included in the study or whether the research ethics committee or IRB specifically waived the need for their consent."

4. Please include a complete copy of PLOS’ questionnaire on inclusivity in global research in your revised manuscript. Our policy for research in this area aims to improve transparency in the reporting of research performed outside of researchers’ own country or community. The policy applies to researchers who have travelled to a different country to conduct research, research with Indigenous populations or their lands, and research on cultural artefacts. The questionnaire can also be requested at the journal’s discretion for any other submissions, even if these conditions are not met.  Please find more information on the policy and a link to download a blank copy of the questionnaire here: https://journals.plos.org/plosone/s/best-practices-in-research-reporting. Please upload a completed version of your questionnaire as Supporting Information when you resubmit your manuscript.

“This study was not funded by a grant. It is a PhD project and was supported by the Strategic and Collaborative Capacity Development in Ethiopia and Africa (SACCADE) Project, Norwegian Program for Capacity Development in Higher Education and Research for Development (NORHAD), University of Oslo.”

6. We note that Figure 1 in your submission contain [map/satellite] images which may be copyrighted. All PLOS content is published under the Creative Commons Attribution License (CC BY 4.0), which means that the manuscript, images, and Supporting Information files will be freely available online, and any third party is permitted to access, download, copy, distribute, and use these materials in any way, even commercially, with proper attribution. For these reasons, we cannot publish previously copyrighted maps or satellite images created using proprietary data, such as Google software (Google Maps, Street View, and Earth). For more information, see our copyright guidelines: http://journals.plos.org/plosone/s/licenses-and-copyright.

Reviewers' comments:

Reviewer's Responses to Questions

**Comments to the Author**

1. Is the manuscript technically sound, and do the data support the conclusions?

Reviewer #1: Yes

Reviewer #2: Partly

2. Has the statistical analysis been performed appropriately and rigorously? 

Reviewer #1: Yes

Reviewer #2: Yes

3. Have the authors made all data underlying the findings in their manuscript fully available?

Reviewer #1: Yes

Reviewer #2: Yes

4. Is the manuscript presented in an intelligible fashion and written in standard English?

Reviewer #1: Yes

Reviewer #2: No

5. Review Comments to the Author

Reviewer #1: Authors have performed on TB research by focusing on delay: please see as attached

Title

Why authors are interested to total delay… as it did not indicate which delay was important in this research… that is delay related to patients or delay related to health care system?

In abstract and end of backgrounds… the aim of this study is not clearly indicated, these also reflected in results and discussion parts … that with open discussion… It would be better if the aim was clear and others were aligned to it.

In methods

In line 152, total defined as “the period between onset of TB symptoms and the first start of anti-TB treatment”, in this statement onset of TB symptoms... did it mean five TB symptoms onset (Cough, fever, weight loss, night sweating and Haemoptysis )

Results

In line 153, the reason why the median total delay was taken for the classification of delayed or not was not clear

In Table 1… there was “Town”… but in 318… authors used “Urban”… which one is correct? Can we use the terms interchangeable?

From 113 to 114 and in Table 1…authors used the term community based and facility based DOTS, but these terms were not defined

In Table 3, the multivariable analysis for the determinants of delay was indicated, in the Table many variables were indicated? Did that all variables or model run at once or any else? It was not clear indicated in methods part

Discussion

Starting from line of 316, how the authors relate the patients followed facility based DOTS were less delayed when compared with community Based DOTS… but in other study published on BMJ open (https://bmjopen.bmj.com/content/bmjopen/11/7/e048369.full.pdf) with the same data… CBD was more effective than FBD?

Conclusion

In line 350, authors indicted as unacceptable long total delay …. Better if rephrased as there is no acceptable total delay

From line 350-353… the conclusion has no results background

Reference

Despite the literatures are available, it would be better if selective

Reviewer #2: Please find the attached PDF. Despite the author present the major public health problem in Ethiopia, it needs revision before publication. The author shall report related to patient delay and health system delay. What will its implication to the program.

6. PLOS authors have the option to publish the peer review history of their article (what does this mean?). If published, this will include your full peer review and any attached files.

Reviewer #1: **Yes: **Hussen Mohammed

Reviewer #2: No

---

## [Author Response · Author response to Decision Letter 0]

7 Oct 2022

PONE-D-22-02817

Total delay and associated factors among tuberculosis patients in Jimma Zone, Southwest Ethiopia

Response to the academic editor

Dear Dr.Wenhui Mao (Academic Editor of PLOS ONE)

Thank you so much for your insightful comments and we are glad for the opportunity given to submit the revised version of the manuscript. We have modified the manuscript in response to your valuable comments. We hope that the revised version of our manuscript will still be considered by PLOS ONE. Please see below is a point-by-point response to the comments raised by the editor and reviewers. 

Kind regards, 

S.No Comments raised by academic editor Authors’ response 

1 Please ensure that your manuscript meets PLOS ONE's style requirements, including those for file naming. Comment accepted. We have revised the manuscript based on the PLOSE ONE journal style template and journal requirements. 

2 We note that this work is related to the following submission of yours currently under review: PONE-D-22-05873 "Determinants of unfavorable treatment outcome among tuberculosis patients in Jimma Zone, Southwest Ethiopia". Please ensure you have discussed the related submission in your cover letter. Please also revise your Baseline Demographics table to ensure that only clinical characteristics relevant to the current study are included, to avoid inappropriate overlap between the two works.

Comment accepted. PONE-D-22-05873 "Determinants of unfavorable treatment outcome among tuberculosis patients in Jimma Zone, Southwest Ethiopia" is not now under review. The editor has rejected it by considering some overlapping with the current study. 

3 You indicated that you had ethical approval for your study. In your Methods section, please ensure you have also stated whether you obtained consent from parents or guardians of the minors included in the study or whether the research ethics committee or IRB specifically waived the need for their consent 

Comment accepted. For minors (15 – 17years), assent was obtained from themselves and consent was secured from their parents or guardians. 

Text included under ethical consideration subsection page 8, lines 196 and 197. 

4 Please include a complete copy of PLOS’ questionnaire on inclusivity in global research in your revised manuscript. Our policy for research in this area aims to improve transparency in the reporting of research performed outside of researchers’ own country or community. The policy applies to researchers who have travelled to a different country to conduct research, research with Indigenous populations or their lands, and research on cultural artefacts 

The current study was not performed outside of the researchers’ own country or community. Because, it was conducted in the first author’s (corresponding author’s) country and community. We think that it is not necessary to include PLOS’ questionnaire on inclusivity in global research in our revised manuscript. If you still think it is necessary, we can include it. 

5 Please provide an amended statement that declares *all* the funding or sources of support (whether external or internal to your organization) received during this study, as detailed online in our guide for authors at http://journals.plos.org/plosone/s/submit-now. Please also include the statement “There was no additional external funding received for this study.” in your updated Funding Statement.

Comment accepted. We have included your comments and updated our funding statement. In addition, the amended Funding Statement has been mentioned in our cover letter. 

6 We note that you have indicated that data from this study are available upon request. PLOS only allows data to be available upon request if there are legal or ethical restrictions on sharing data publicly. In your revised cover letter, please address the following prompts:

If there are no restrictions, please upload the minimal anonymized data set necessary to replicate your study findings as either Supporting Information files or to a stable, public repository and provide us with the relevant URLs, DOIs, or accession numbers.

Comment accepted. We are sincerely sorry for not including dataset in the previous submission but now we have included the data set as supportive information file (S1_ Data). 

7 We note that Figure 1 in your submission contain [map/satellite] images which may be copyrighted. All PLOS content is published under the Creative Commons Attribution License (CC BY 4.0), which means that the manuscript, images, and Supporting Information files will be freely available online, and any third party is permitted to access, download, copy, distribute, and use these materials in any way, even commercially, with proper attribution. For these reasons, we cannot publish previously copyrighted maps or satellite images created using proprietary data, such as Google software. 

Comment accepted. We have removed figure 1 from our submission. 

Response to Reviewer #1

Dear Dr. Hussen Mohammed

Thank you so much for your constructive review and valuable comments. 

We have modified the manuscript in response to your extensive and insightful comments. A detailed point-by-point response to your comments and questions is presented below: 

Kind regards, 

S.No Comments raised by Reviewer #1 Authors’ response 

1 Title 

Why authors are interested to total delay… as it did not indicate which delay was important in this research… that is delay related to patients or delay related to health care system? 

 Comment accepted. As can be observed in the result section of this manuscript, the median patient delay was 30 days which is relatively acceptable. Several previous studies considered a median patient delay of 30 days as acceptable and used this figure to dichotomize the sample into delay and non-delay periods. In addition, the health systems delay is very short as you can see in the result section. Therefore, we wanted to analyze the total delay (which is relatively longer) and assess how total delay affects the diagnosis and treatment start among TB patients. Based on your valuable comment, we have included a text about the magnitude of delay in TB diagnosis and treatment, about patient delay and health system delay in the revised manuscript page 3, lines 69-76. 

2 In abstract and end of backgrounds… the aim of this study is not clearly indicated, these also reflected in results and discussion parts … that with open discussion… It would be better if the aim was clear and others were aligned to it. 

 Comment accepted. The aim of the current study is now stated as “the aim of the present study was to determine the length of total delay and analyze factors associated with total delay”. It has been included in the abstract, introduction, results, and discussion sections. Please see, on page 2, lines 37 and 38; page 4, lines 102 and 103; page 8, line 200; page 14, line 259

3 In methods 

In line 152, total defined as “the period between onset of TB symptoms and the first start of anti-TB treatment”, in this statement onset of TB symptoms... did it mean five TB symptoms onset (Cough, fever, weight loss, night sweating and Haemoptysis )

 Comment accepted. First, we asked patients which TB symptoms they have been suffering from (including the mean five TB symptoms onset). Then, we asked duration between onset of the above symptoms until their first visit to a medical provider, duration from first visit to a medical provider until diagnosis of TB, and duration from DX of TB until first treatment started. Finally, we got total delay by adding the above three durations. Reported on page 12, table 3. 

4 Results 

In line 153, the reason why the median total delay was taken for the classification of delayed or not was not clear. 

 Comment accepted. We took median total delay than mean for classification since our data was not normally distributed. As you know, mean is affected by extreme values. Therefore, it is common to use median if the data is not normally distributed.

5 In Table 1… there was “Town”… but in 318… authors used “Urban”… which one is correct? Can we use the terms interchangeable? Comment accepted. 

Sorry for the inconsistency. Now, we have corrected it as urban on page 9, table 1. 

6 From 113 to 114 and in Table 1…authors used the term community based and facility based DOTS, but these terms were not defined. 

 Comment accepted. Thank you so much for your careful review. We have included these terms under operational definitions subsection, page 7, lines 171-174. 

7 In Table 3, the Multivariable analysis for the determinants of delay was indicated; in the Table many variables were indicated? Did that all variables or model run at once or any else? It was not clear indicated in methods part 

Candidate variables in the bivariable analysis (p-value of < 0.25) were included for multivariable analysis with a backward stepwise method. Please see page 6, line 161. 

8 Discussion

Starting from line of 316, how the authors relate the patients followed facility based DOTS were less delayed when compared with community Based DOTS… but in other study published on BMJ open (https://bmjopen.bmj.com/content/bmjopen/11/7/e048369.full.pdf) with the same data… CBD was more effective than FBD?

 The previous study compared TB treatment outcomes among a cohort of TB patients attending CB-DOTS versus FB-DOTS. The finding showed that the CB-DOTS approach was a better performance in terms of treatment outcomes (cure rate, treatment failure rate, and sputum conversion rate). Whereas, the present study showed patients under FB-DOTS were less delayed than patients under CB-DOTS. In this data, there was no statistically significant association between total delay and treatment outcomes (AOR= 0.76, 95% CI 0.40 -1.42 and P. value = 0.386). This may be the reason for the difference. 

9 Conclusion 

In line 350, authors indicted as unacceptable long total delay …. Better if rephrased as there is no acceptable total delay

 Comment accepted. The term was clarified on page 19, line 382. 

10 From line 350-353… the conclusion has no results background 

 Comment accepted. We have included the result such as the median total delay of 35 days. Factors associated with total delay such as poor knowledge of TB, swelling or wound around the neck region, being women, and low level of education are also mentioned in the conclusion section page 19, lines 382 – 385. 

11 Reference 

Despite the literatures are available, it would be better if selective 

 Comment accepted. Based on the reviewers’ comments, some of the references were removed and some of the references were included. 

Response to Reviewer #2

Dear Reviewer, 

Many thanks for your constructive review and valuable comments. 

We have modified the manuscript in response to your extensive and insightful comments. A detailed point-by-point response to your comment is presented as follow: 

Kind regards, 

S.No Comments raised by Reviewer #2 Authors’ response 

1 Title: Why total delay? Patient delay and health system delays are not explicitly mentioned in your report. Does it gives sense for programmers? In which area the gap happens, in the health system in TB diagnosis and treatment; or patients. Comment accepted. 

As can be observed in the result section of this manuscript, the median patient delay was 30 days which is relatively acceptable. Several previous studies considered a median patient delay of 30 days as acceptable and used this figure to dichotomize the sample into delay and non-delay periods. In addition, the health systems delay is very short as you can see in the result section. Therefore, we wanted to analyze the total delay (which is relatively longer) and assess how total delay affects the diagnosis and treatment start among TB patients.

Based on your insightful comment, we have reported the patients, health system, and total delays in the result section. Page 11, lines 232-234 and page 12, table 3. 

2 Abstract 

Prospectively. This might confuse readers. This is a cross-sectional study and should use appropriate word. Or clarify why does it mean in your context?

Unacceptably. what was your comparison? or from your personal judgment? 

Comment accepted. These terms have been removed. 

3 Introduction: The introduction section does not mention about patient, health system delay. As far as delay is concerned, these concepts are critical, otherwise it will not have any clinical implication Comment accepted. We have included a text about patient and health systems delay in the introduction section page 3, lines 69-76. 

4 Line 71: which indicator? Nice to include in your manuscript Comment accepted. We have included the targets set for the End TB Strategy such as to decrease TB mortality by 95% and TB incidence by 90% by 2035 (baseline of 2015), and to certify that no family is suffered with TB related catastrophic costs, page 3, lines 78- 80.

5 Line 79: which study with the lowest, and which highest? The lowest 30 days (Zimbabwe and Vietnam) and the highest 366.5 days in Afghanistan include on page 4, line 88. 

6 Line 81: reference

Line 84: put reference in each sections Comment accepted. The references are included for each on page 4, lines 92-96

7 Line 86: what does it means? there are various studies in Ethiopia regarding TB delay in diagnosis and treatment. Comment accepted. By that we mean there are various diagnostic and treatment delay studies conducted in various parts (regions) of Ethiopia. However, there is limited delay study in the study area (in Jimma Zone)

8 Line 87: Do you have any evidence for this claim? Yes, we have included a references on page 4, line 101. 

9 Good to include the number of TB patients by type in the study setting. Comment accepted. We have included the number of TB patients in the revised manuscript page 5, lines 114 and 115.

10 Line 108: What does it mean? it makes confusion. What is the sampling technique for this study?

Line 109: What does it mean? Comment accepted. We have modified the wording as “A total of 1,161 newly diagnosed TB patients were included from sampled districts’ public health facilities and health posts”, please see page 5, lines 116 and 117.

11 Line 117: why you take 50% while there are studies?

Line 119: But in the above statement (study population and sampling), you mentioned all included???? Comment accepted. Yes, we could have used previous study data instead of considering the 50% proportion which is applied when there is no previous data on the proportion of delay. We used the 50% proportion in the formula to just show the required minimum sample size for addressing the research question. As we have included all (1,161) participants which is more than the required minimum sample size for this study, we are confident that the sample size represents the population. In the revised manuscript we have included the following text. “The inclusion of the 1,161 study participants in the current study is advantageous as it provides more than adequate representative population to address the research questions”, page 5, lines 130 and 133. 

12 Line 146: Bivariate, what it means? Bivariable

Line 147: Multivariate vs multivariable? Comment accepted. We have modified the terms as Multivariable and bivariable on page 6, lines 159-161.

13 Line 152: where is patient delay, health system delay? Patient and health system delays were included under operational definitions subsection, page 7, lines 165 and 166.

14 Line 153: Any evidence for this definition? Operationally defined as delayed if the period between the onset of TB symptoms and first start of anti-TB treatment is more than the calculated median total delay (> 35 days) on page 7, lines 169 and 170. 

15 why you take both oral and written consent?

 Comment accepted. This has been clarified in the revised manuscript as “written (for literate) and oral (for illiterate) informed consent was secured from the study participants before starting the data collection”, page 8, lines 194 and 195.

16 Line 178: It is not a cohort study. Do not use this term for this study, otherwise it will confuse the reader. Comment accepted. We have modified the wording as “…a total of 1,161 patients participated in this study”, page 8, line 201.

17 Table 1 HIV status: where do you get this data? Comment accepted. We have clarified this as “Clinical data such as HIV status, main mode of TB diagnosis, TB classification were collected by a checklist attached to the questionnaire from TB register”, page 6, lines 146 and 147. 

18 Table 3: Have you checked collinearity? If that is the case, what was the finding? There are variable that possibly correlated. 

 Yes, we have checked the variables for collinearity problem and the result of VIF (Variance Inflation Factor) for the variables were less than 2 except that occupation of respondent and source of household income had VIP of 3.955 and 3.928, respectively. Besides, we used a large sample size and multicollinearity is less likely to be a problem in this study. 

19 Line 245: How do check it? Have you done any analysis? Comment accepted. We have removed the sentence since it was already mentioned as “A former study in Ethiopia revealed that individuals with a higher level of education had better knowledge about TB compared to those who did not read and write” Please see page 15, lines 284 and 285. 

20 Line 248: What does causative factor means? It is hard to talk about causation in this study, the scope of this study might limit for this concept. Comment accepted. We have modified the sentence as “Our study showed that the educational level of the patients may not be directly associated with the delay but do significantly affect the knowledge level of the study participants” on page 15, lines 285 and 287. 

21 Line 277: Do you mean perceived stigma is higher in females as compared to males? what cultural belief in female that makes delay? We have clarified as “In terms of cultural beliefs, women in Ethiopia give priority for their family than themselves, have main responsibility for children care, have less decision making power, and worries about preserving modesty might affect their health care seeking behavior. Previous study revealed that women were more likely to suffer from TB stigma compared to men” page 16, lines 315-318.

 Many thanks for your valuable questions and comments.

---

## [Decision Letter · Decision Letter 1]

26 Jan 2023

Total delay and associated factors among tuberculosis patients in Jimma Zone, Southwest Ethiopia

PONE-D-22-02817R1

Dear Dr. Ereso,

We’re pleased to inform you that your manuscript has been judged scientifically suitable for publication and will be formally accepted for publication once it meets all outstanding technical requirements.

Kind regards,

Wenhui Mao, PhD

Academic Editor

PLOS ONE

Additional Editor Comments (optional):

Reviewers' comments:

Reviewer's Responses to Questions

**Comments to the Author**

1. If the authors have adequately addressed your comments raised in a previous round of review and you feel that this manuscript is now acceptable for publication, you may indicate that here to bypass the “Comments to the Author” section, enter your conflict of interest statement in the “Confidential to Editor” section, and submit your "Accept" recommendation.

Reviewer #1: All comments have been addressed

2. Is the manuscript technically sound, and do the data support the conclusions?

Reviewer #1: Yes

3. Has the statistical analysis been performed appropriately and rigorously? 

Reviewer #1: Yes

4. Have the authors made all data underlying the findings in their manuscript fully available?

Reviewer #1: Yes

5. Is the manuscript presented in an intelligible fashion and written in standard English?

Reviewer #1: Yes

6. Review Comments to the Author

Reviewer #1: I thanks the authors for revising their manuscript

In abstract line 41, change town to urban as in the body of manuscript

On page 8, line 203, the authors indicated that as there were written consent for literate and verbal consent for illiterate, in consent if we had different study populations that can write and read, and for illiterate, witness is recommended that read for them but here different, it would be better if described further as either verbal consent with document or verbal consent without document for illiterate (39.5%)

7. PLOS authors have the option to publish the peer review history of their article (what does this mean?). If published, this will include your full peer review and any attached files.

Reviewer #1: **Yes: **Hussen Mohammed (PhD), Public Health Department, Dire Dawa University, Ethiopia

---

## [Editor Report · Acceptance letter]

30 Jan 2023

PONE-D-22-02817R1 

Total delay and associated factors among tuberculosis patients in Jimma Zone, Southwest Ethiopia 

Dear Dr. Ereso:

I'm pleased to inform you that your manuscript has been deemed suitable for publication in PLOS ONE. Congratulations! Your manuscript is now with our production department. 

Kind regards, 

on behalf of

Dr. Wenhui Mao 

Academic Editor

PLOS ONE